# Improving the Abrasion Resistance of Nodular Cast Iron Castings by Remelting Their Surfaces by Laser Beam

**DOI:** 10.3390/ma17092095

**Published:** 2024-04-29

**Authors:** Tomasz Wróbel, Andrzej Studnicki, Marcin Stawarz, Czesław Baron, Jan Jezierski, Dariusz Bartocha, Rafał Dojka, Jacek Opiela, Aleksander Lisiecki

**Affiliations:** 1Department of Foundry Engineering, Silesian University of Technology, 7 Towarowa Street, 44-100 Gliwice, Poland; andrzej.studnicki@polsl.pl (A.S.); marcin.stawarz@polsl.pl (M.S.); czeslaw.baron@polsl.pl (C.B.); jan.jezierski@polsl.pl (J.J.); dariusz.bartocha@polsl.pl (D.B.); 2Odlewnia, “RAFAMET” Sp. z o.o., 1 Staszica Street, 47-420 Kuźnia Raciborska, Poland; r.dojka@odlewnia-rafamet.pl (R.D.); j.opiela@odlewnia-rafamet.pl (J.O.); 3Department of Welding Engineering, Silesian University of Technology, 18a Konarskiego Street, 44-100 Gliwice, Poland; aleksander.lisiecki@polsl.pl

**Keywords:** surface hardening, laser remelting, nodular cast iron, abrasive wear

## Abstract

This paper presents the results of research conducted in the field of the technology of surface hardening of castings from unalloyed and low-alloy nodular cast iron using the laser remelting method. The range of studies included macro- and microhardness measurements using Rockwell and Vickers methods as well as metallographic microscopic examinations using a scanning electron microscope. Moreover, abrasive wear resistance tests were performed using the pin-on-disk method in the friction pair of nodular cast iron—SiC abrasive paper and the reciprocating method in the friction pair of nodular cast iron—unalloyed steel. Analysis of the test results shows that the casting surface layer remelting by laser for unalloyed nodular cast iron results in a greater improvement in its resistance to abrasive wear in the metal–mineral system, as compared to low-alloy cast iron. Additionally, carrying out the laser hardening treatment of the surface layer made of the tested grades of nodular cast iron is justified only if the tribological system of the cooperating working parts and allowable dimensional changes during their operation are known.

## 1. Introduction

Nodular cast iron belongs to the group of engineering materials that feature good mechanical properties, providing designers with great application opportunities for this material. Rich and very diverse standardized grades of nodular cast iron offer a whole range of mechanical properties, the proper selection of which determines the safety and durability of machine components under design [1,2,3,4,5,6,7,8,9,10]. The working parts of machines during the period of their operation are often exposed to abrasive wear, most frequently affecting the outer surface layer of the part material [11,12]. Materials engineering in this scope is looking for methods to protect the surface of the working part against the effects of interactions with cooperating components in various tribological systems. In tribological systems of metal–ceramic grains, there is usually very intensive wear of the metal component, and its volumetric wear reaches deep into the material. The intensity of this abrasive wear can be reduced but cannot be fully eliminated. The decisive parameter here is the hardness of the working surface being in contact with very hard ceramic grains. For the tribological system under consideration, there is a certain regularity that defines the component as weaker in terms of resistance to abrasive wear, and this component is the one with a lower working surface hardness. Any actions eliminating or reducing the difference in the hardness of cooperating parts in a tribological system may result in extending the life of the parts [10,11,12,13,14,15]. 

The effect of the material being subjected to abrasion on abrasive wear has been confirmed by many researchers. In general, it can be concluded that the increase in hardness of the wearable parts results in increasing the resistance to abrasive wear. The research on the relationship between the hardness of the wearable material and the abrasive material is very interesting, as shown in publication [14]. Experiments covering many materials have shown that their resistance to abrasive wear is additive; i.e., it is a result of adding up the resistance of individual structural components. Examples of the conclusions resulting from the abrasion resistance of selected steels, rubbed against a corundum abrasive cloth, have shown the following regularities: pearlite, sorbit and troostite abrasion resistance depends on cementite dispersion, while the martensite abrasion resistance is greater according to the amount of carbon it contains, and as the residual austenite in the martensitic structure increases, the abrasion resistance decreases. Lamellar pearlite has a greater resistance to abrasive wear than spherical pearlite. For unalloyed cast iron, the resistance to abrasive wear depends on the shape and dispersion of graphite and the share and the form of phosphorus eutectic. The properties of the listed material phase components emphasize the role played by the parameters of the metallographic structure in the course of the abrasive wear process. It is the operation environment of specific parts that has a significant impact on their abrasive wear. It is associated with the presence of impurities, their oxidizing properties and moisture. The load on the tribological system is one of the most important factors determining the intensity of abrasive wear. Abrasive wear is directly proportional to the pressure per unit area and friction distance. 

In connection with the problems mentioned above, this paper describes activities aimed at improving the resistance to abrasive wear in specific tribological systems of selected nodular iron grades after laser remelting of the casting surface layer. This process results in changing the structure of the casting surface layer, reaching a depth of several millimeters. This hardening method of cast iron castings’ surface layer is also presented in papers [16,17,18]. However, in comparison to the papers mentioned above, the authors proposed more detailed studies among other things in the range of tribological systems. Therefore, the aim of the studies was to determine the effect of laser remelting the surface layer of castings made of low-alloy and unalloyed nodular cast iron on wear resistance. 

## 2. Experimental Methods

Table 1 presents the characteristics of the low-alloy nodular cast iron (melt marked as W1 and W2) and unalloyed nodular cast iron (melt marked as W3 and W4) under examination. 

The exact chemical compositions of cast iron are not provided. They are part of the knowledge that is the secret of the foundry—the beneficiary of the project under which they were developed. The tests were performed using four types of cast iron; W1 and W3 were nodular cast iron with increased elongation values (Polish patent pending P.443934), and W2 and W4 were nodular cast iron with increased strength (Polish patent pending P.443935).

However, the ranges of the element concentrations in the studied nodular cast iron W1–W4 are given below: Gray cast iron GJS-700-4 grade contains the following: 0.30 to 0.50 wt.% Mn, 0.03 to 0.08 wt.% Mg, less than 0.05 wt.% Cr and less than 0.01 wt.% V. It is characterized by additionally containing the following: 3.10 to 3.45 wt.% C, 2.35 to 2.75 wt.% Si, less than 1.10 wt.% Ni, less than 0.50 wt.% Mo and 0.40 to 0.70 wt.% Cu.Gray cast iron GJS-750-2 grade contains the following: 0.35 to 0.55 wt.% Mn, 0.03 to 0.08 wt.% Mg, less than 0.05 wt.% Cr and less than 0.01 wt.% V. It is characterized by additionally containing the following: 3.20 to 3.45 wt.% C, 2.35 to 2.65 wt.% Si, 0.7 to 1.50 wt.% Ni, 0.30 to 0.50 wt.% Mo and 0.40 to 0.70 wt.% Cu.Gray cast iron GJS-600-6 grade contains the following: 0.30 to 0.50 wt.% Mn, 0.03 to 0.08 wt.% Mg, less than 0.05 wt.% Cr, less than 0.01 wt.% V, less than 0.05 wt.% Ni and less than 0.01 wt.% Mo. It is characterized by additionally containing the following: 3.10 to 3.45 wt.% C, 2.35 to 2.75 wt.% Si and 0.40 to 0.70 wt.% Cu.Gray cast iron GJS-650-3 grade contains the following: 0.35 to 0.55 wt.% Mn, 0.03 to 0.08 wt.% Mg, less than 0.05 wt.% Cr, less than 0.01 wt.% V, less than 0.05 wt.% Ni and less than 0.01 wt.% Mo. It is characterized by additionally containing the following: 3.20 to 3.45 wt.% C, 2.35 to 2.65 wt.% Si and 0.40 to 0.70 wt.% Cu.

The range of studies included the following:The preparation and the execution of melts containing selected types of nodular cast iron;The preparation of test samples from “Y” type castings (Figure 1);Carrying out the operation of laser remelting for selected surfaces of test samples (Figure 1) after completing the preliminary tests to select the laser operating parameters;Performing metallographic tests of the laser-remelted surface layer of the samples;Performing abrasive wear tests in two tribological systems (1st metal-fixed ceramic grains, 2nd metal-to-metal).

To modify the surface layer of the samples, the process of rapid remelting and crystallization of the thin near-surface layer of the casting by the laser technique was used. The extreme conditions of this process resulted in obtaining a different structure of the remelted layer and the adjacent layer subjected to the heat of the melting process and rapid cooling. To properly implement the process of laser remelting of the nodular cast iron under test, a series of preliminary tests were carried out first to estimate the range of variability for laser operating parameters, and then the actual tests were performed. Generally, in studies, the laser power 800–3300 W was used. Moreover, the position of the laser focus relative to the top surface was determined for all the tests to 35 mm; the diameter of the argon blowing nozzle was 15 mm, and its volumetric flow rate was 15 L/min. Subsequent laser “stitches” were applied with offsets of 4 mm, 3 mm, 2.5 mm and 2 mm to assess the effect of overheating intensity on the results obtained. All the studies were conducted using a TRUMPF TruDisk 3302 disc laser (TRUMPF, Ditzingen, Germany), as shown in Figure 2.

To assess the effects of laser treatment, measurements of hardness directly (without machining) on the remelted surface using Rockwell’s method were carried out. Hardness was measured by applying the SUNPOC SBRV-100D universal tester (Guizhou Sunpoc Tech Industry Co., Ltd., Guiyang City, China) using a diamond cone intender with an apical angle of 120° loaded with a force of 1.470 N. For each sample, 10 measurements were taken, moving the penetrator across the sample, perpendicularly to the “stitches”.

Additionally, the measurements of microhardness on the cross-sections of the samples were carried out. The measurements were carried out using Vickers’s method and a FUTURE-TECH FM 700 tester (FUTURE-TECH CORP., Fujisaki, Japan) by applying a diamond correct pyramid intender with an apical angle of 136° and loaded with a force of 4.9 N at time 10 s.

The microstructure research was conducted using a Phenom ProX scanning electron microscope (Phenome-World, Eindhoven, Netherlands) (SEM) with backscattered electron (BSE) imaging and electron beam accelerating voltages of 5 and 10 kV.

Abrasive wear can occur in many tribological systems that can be found in machines. Working parts, cooperating with each other directly or through an intermediary medium, are particularly exposed to wear. The article presents the results of abrasive wear tests in two different tribological systems. In the system composed of metal (abraded sample of nodular cast iron) and fixed abrasive grains (the counter-sample was an abrasive paper with a specific ceramic grit size—SiC), there was a significant difference in the hardness of these system components. In the second system under analysis, composed of metal (an abraded sample of nodular cast iron) and metal (the counter-sample was a steel sheet—S235 grade), the difference in hardness was in favor of the abraded sample. For both abrasive wear test cases, a sample of geometrical dimensions, shown in Figure 3, was used. This sample was cut out from the test sample after completing the laser remelting process.

In the first system, i.e., the metal—abrasive paper system, a stand developed in the Department of Foundry Engineering, the Silesian University of Technology, was used to test the abrasive wear. The stand is largely modeled on the solutions of the pin-on-disc test methods, where the pin-shaped test sample is pressed against a rotating disk, causing abrasive wear. In the Tribotester 3-POD (SUT, Gliwice, Poland), 3 samples are simultaneously pressed against the rotating disk. The samples are also rotated in a special holder. Figure 4 shows the operation diagram of this machine and its general view. This machine is used to test three samples at the same time. The samples are mounted in a special rotating holder, which allows users to set an individual pressure to the abrasive disc for each sample. The abrasive disc is a wheel of sandpaper with a specific grit size and type of abrasive grain. A new abrasive disc was used for each set of samples. The rotation direction of the abrasive disc is opposite to that of the sample holder. The adjustable operating parameters of the machine include the rotations of the abrasive disc, rotations of the sample holder, pressure of the samples, type of the abrasive disc and experiment duration.

The following operating parameters were adopted for the Tribotester 3-POD in the tests performed:Rectangular prism sample with dimensions 10 × 15 (abraded surface) × 25 mm;Abrasive disc: abrasive paper 80, SiC;Abrasive disc revs (counter-sample): 170 rpm;Abraded sample clamp revs: 300 rpm;Loading per single sample: 230 g;Abrasion path per cycle: 800 m;Total abrasion path: 4800 m;Dry abrasion;Sample dimensions 10 × 15 × 25 mm;Room temperature.

The weight loss was measured after each of the 6 measurement cycles (total distance traveled by the sample, 6 × 800 m = 4800 m) using scales with an accuracy of 0.001 g. 

As a part of this study, the Tribotester 3-POD stand was used to carry out abrasion tests of the nodular cast iron in the as-cast state and after various treatments of refining the surface layer of the casting under test. The model (reference) material was low-alloy steel, resistant to abrasive wear with the commercial name CREUSABRO^®^8000 (marked with C8) and with 45 HRC hardness. Such a reference material was adopted for all the abrasive wear tests performed on the Tribotester 3-POD measuring stand in the Department of Foundry Engineering, at the Silesian University of Technology.

In the second system, i.e., the metal-to-metal system, another tribological machine was used in the tests. It consists of a drive part composed of an electric motor, belt transmission, toothed transmission and crank mechanism. The working part includes a jaw chuck that performs a reciprocating (horizontal) movement and a table that can move vertically, on which a static force is applied by a lever system. Figure 5 shows the stand’s kinematic diagram and the general view of the machine. 

All the samples were mounted in the holder of the tribological machine in the same way. Performing one measurement cycle consisted in measuring the weight loss of the tested material after performing 1500 cycles of the associated system’s reciprocating motion. Each sample covered a distance of 150 m during the tests. A steel plate grade of S235 was used as a counter-sample. The sample load was 50 N. Before the tests, all the samples were lapped to ensure full contact between the samples and the counter-sample. The weight loss was measured after a full test cycle (the distance covered by the sample was 150 m) using scales with an accuracy of 0.001 g. Before each test cycle, the scales were checked with mass standards. Before each test cycle of the sample, the scale was checked and calibrated with mass standards. The measurement error was ±0.001 g. The air temperature in the laboratory during the tests oscillated within the limits of 21 ± 0.5 °C.

## 3. Results

### 3.1. Preliminary Tests—Selecting Laser Operating Parameters

The preliminary tests were carried out for the full range of laser power as well as for different (feed) speeds of its beam using example nodular cast iron samples. Table 2 summarizes the present laser parameters together with the obtained surface hardness after laser remelting. Figure 6 shows the views of the selected sample surfaces after laser remelting in preliminary tests. 

The analysis of the hardness measurement results, in association with the laser power value, does not indicate a simple relationship after increasing the value from 800 W to 3300 W. However, in general, increasing the laser power helps to obtain greater hardness. Secondly, it can be seen that the increase in hardness is also related to the laser beam feed rate; for example, for samples P1 to P8, a significant increase in hardness can be seen with the increasing laser power and decreasing feed rate, which results in stronger surface remelting. Whereas, for samples P9 to P14, one can see the trend of increasing hardness with increasing laser power at a constant feed speed of 1000 mm/min (Figure 7). However, from the economic point of view for the laser treatment, it is important to find the best combination of laser power and feed speed to make the process as short as possible and to make subsequent “stitches” before the previous ones cool down completely. Moreover, from the technological point of view, it is very important for the life of laser-treated cast iron castings to search for the most favorable process parameters in terms of obtaining the highest surface hardness.

The views of the remelted surface in a few cases, especially for the highest laser power (Figure 6b), show numerous penetrations and cavities, often large in size. If one compares the appearance of the surface treated with the laser and the surface of the samples prior to this treatment, i.e., after machining, one can see that defects of this type were barely found. This indicates that porosity-type defects (especially the gaseous ones) occur inside the samples, and intense heat application in a laser beam reveals these defects. In addition, it is possible that, in some areas of the laser beam’s application, it burns clusters of graphite that could segregate near some surfaces (the so-called graphite flotation).

### 3.2. Main Tests—Laser Remelting of the Test Sample Surfaces

Based on the preliminary tests, the variability range for the main test parameters was assumed. It was decided that the feed speed would be as high as possible, i.e., 1000 mm/min, while the laser power had three values: 2000, 2500 and 3000 W. Each laser power was used to treat three samples. Figure 8 shows a set of samples after laser remelting for the melt W4. 

Surface hardness measurement of the samples showed a significant increase in hardness (Table 3) as compared to the as-cast state of the cast iron under test. Figure 9 shows hardness results in a graphical form. 

Table 4 summarizes the results of microhardness measurements on the cross-sections of the tested nodular cast iron samples after the laser treatment. The table also provides the thickness of the hardened layer (g, µm) that meets the hardness criterion (>400 HV). Figure 10 shows an example diagram of the microhardness distribution on the cross-section of the tested sample with a layer hardened by laser treatment.

The process of hardening the surface of nodular cast iron casting by laser remelting makes it possible to obtain a relatively thick layer (over 1 mm) with a hardness exceeding 400 HV. The resulting layer consists of two basic zones. 

Zone 1 is the effect of remelting nodular cast iron, which after the melting crystallizes as white cast iron with a ledeburitic structure and a hardness exceeding 900 HV (dendritic ledeburitic structure). Zone 2 is the effect of the thermal environmental impact (heat-affected zone) where the classical heat treatment processes can take place—the structure shows acicular phases (martensite and bainite) against the background of austenite. In the remaining part of the sample’s cross-section, there is a microstructure unchanged in comparison to the as-cast state, i.e., containing nodular graphite in the pearlitic-ferritic matrix in the case of unalloyed cast iron or only in the pearlitic matrix in the case of low-alloy cast iron. The remelted zone shows a fairly stable hardness of approx. 900 HV. Zone 2 shows a change in hardness depending on the distance from the sample surface. It is assumed that this zone ends when the hardness drops below 400 HV. Figure 11 shows an example of the structure in the hardened zones under testing for melt W2 obtained by laser with a power of 2500 W.

Analyzing the fractures on a scanning microscope allowed a certain phenomenon to be observed that occurs in Zone 1 of laser remelting. It was found that probably some of the free carbon in the remelting zone moves up toward the surface of the sample, forming a thin free carbon layer. Figure 12 shows the photos of a fracture that illustrates the phenomenon. The local presence of free carbon is a defect, which decreases both the surface quality and hardness of nodular cast iron hardened by laser remelting. Other defects, such as microcracks, were not found in the studied samples. 

Figure 13 shows the diagrams of abrasive wear in the tribological metal—abrasive paper system in a full test cycle for nodular cast iron melts W1 and W3 in the as-cast state and after laser remelting at different power levels, and Figure 14 shows the summary of the total wear of all the tested nodular cast iron samples and the reference sample. Additionally, Table 5 summarizes the total weight loss of the samples after the laser treatment of the nodular cast iron castings’ surfaces and a reduction in the sample wear after this treatment, as compared to the as-cast state of the sample.

By analyzing the wear diagrams (Figure 13), one can notice a clear improvement in the abrasion resistance of the tested nodular cast iron after laser remelting, regardless of the parameters of this process. The wear is not proportional to the abrasion distance, which is related to the wear of the counter sample during the test. Observing this phenomenon, it can be concluded that the abrasion resistance of the test sample has a significant impact on the durability of the counter sample. The increasing wear resistance of the tested sample significantly “destroys” the counter-sample, which is manifested by a greater curvature of the wear graph along the abrasion path.

A more detailed analysis of the wear diagrams indicates the possibility of determining the optimal laser operating parameters to obtain the highest abrasion resistance of the tested nodular cast iron. The data from the graph in Figure 14 and the results in Table 5 indicate what laser parameters should be adopted to obtain the highest improvement in abrasive wear resistance in relation to the nodular cast iron in an as-cast state. For tested nodular cast iron W1–W4, laser remelting of the surface can increase the abrasion resistance compared to the as-cast state by approx. 50 to 70% (Table 5).

Whereas Table 6 and Figure 15 present the results concerning the abrasive wear in the tribological metal-to-metal system. Table 6 summarizes the total weight loss of the samples after the laser treatment of the nodular cast iron castings’ surfaces and a reduction in the sample wear after this treatment as compared to the as-cast state of the sample. Additionally, Figure 15 shows the summary of total abrasive wear for all the nodular cast iron samples in the as-cast state and after surface laser remelting. 

In the case of a metal–metal friction pair, the weight wear of the samples is significantly lower than for testing in a metal–ceramic friction pair. This is due mainly to the low hardness of the counter-sample, i.e., the ferritic-pearlitic unalloyed steel resulting in other wearing mechanisms. In general, in this case of wear, the laser remelting allowed for an average of approximately 50% improvement in the abrasive wear resistance of the tested ductile cast iron compared to the as-cast state.

## 4. Discussion

Laser treatment in the presented research was used to change the properties of a nodular cast iron surface layer without changing its chemical composition. The technique of surface-layer laser remelting with the pre-set operating parameters was used to obtain layer hardening by the crystallization of specific phases in the non-equilibrium conditions typical for rapid melting and cooling processes. In the heat-affected zone, directly adjacent to the remelting zone, ordinary heat treatment processes (e.g., hardening) are also present, which also results in layer hardening. The hardened layer thickness is, therefore, the sum of the thickness of the remelted layer and the heat-treated layer. 

The metallographic examination of the layer hardened by laser remelting allowed us to observe certain phenomena during the cooling process of the nodular cast iron layer remelted with a laser beam. The layer where the liquidus temperature of the cast iron was exceeded was subject to crystallization from the liquid state during the cooling process in unstable conditions. In this layer, a dendritic ledeburitic microstructure with a fine-grained dispersion crystallizes. The layer contains cementite with a high microhardness and can contribute to slowing down the abrasive wear processes. In the remelted layer, free carbon sometimes appears on the upper surface, which is not favorable due to its low hardness and can be easily removed by hard abrasive grains. To remove its residues from the remelted layer, it is necessary to properly manage the remelting process to dissolve it in the crystallizing cast iron.

In addition to the remelted zone in the hardened layer, there is also a zone where ordinary heat treatment takes place, resulting in the formation of acicular martensitic and a bainitic microstructure. These structural components also improve the abrasive wear resistance.

Analyzing the results of abrasive wear in the metal–mineral tribological system shows that the process of remelting the casting surface layer with a laser significantly improves the abrasion resistance of the tested nodular cast iron, especially for unalloyed nodular cast iron. For unalloyed cast irons, in each case, the abrasive wear resistance was enhanced by 60%. It can also be noticed that the thickness of the hardened layer generated depends on the laser power. In each conducted laser remelting trial, where the highest laser power (3300 W) was used, the thickest layers meeting the criterion >400 HV were obtained. 

Moreover, it should be added that the hardness of the surface layers obtained for the nodular cast iron under testing using the laser-melting method increases the abrasion resistance in the metal–ceramic grain combination (the hardness of the ceramic grains used in the abrasion test is approx. 9.2 on the Mohs hardness scale). In the case of low-alloy nodular cast iron, the improvement in abrasion resistance equaled approx. 50%, and in the case of unalloyed nodular cast iron, it equaled approx. 70%, in comparison to the as-cast state. In this case, the thickness of the resulting hardened layer plays an important role in slowing down the wear process, but of course, it does not eliminate it completely. Increasingly higher resistance to abrasive wear should be expected for the lower level of abrasive medium hardness. Moreover, in general, it was concluded that the wear in the case of the metal—abrasive paper friction pair occurs according to the mechanisms of microcutting, scratching and grooving by the effect of SiC grains on the sample surface. 

The abrasive wear tests in the metal-to-metal tribological system of the nodular cast iron under test, hardened by laser remelting of its working surface, have shown the low wear of the test sample. In this case, the hardness of the counter-sample (ferritic-pearlitic unalloyed steel S235 with a hardness of about 200 HB) used in the tests was much lower than the hardness of the layers under analysis, which resulted in a positive test result. A slow and small weight loss of the samples was obtained, which resulted in an increase in resistance to abrasive wear of approx. 50%, as compared to nodular cast iron in the as-cast state. The relatively small weight losses of the samples result in small volume losses, which may be of importance when the laser is applied to refine the working layer of the machine’s parts in metal-to-metal pairs. In this case, i.e., in the metal—metal friction pair, the wear is in proportion to the mechanism of ridging by roughness shearing on the sample and counter-sample surfaces.

## 5. Conclusions

Based on the tests performed and the conducted discussion of the obtained results, the following conclusions were drawn:Carrying out the laser hardening treatment of the surface layer of the working part made of the nodular cast iron under test is justified only if the tribological system of the cooperating working parts and allowable dimensional changes during their operation are known.The scope of the tests performed has shown that the casting surface layer remelting by laser for unalloyed nodular cast iron results in a greater improvement in its resistance to abrasive wear in the metal–mineral system as compared to low-alloy cast iron.The optimum selection of laser operating parameters for remelting the surface layers of the working parts is important to obtain a layer with the desired geometrical characteristics (thickness) and phase composition.The method of laser remelting of the surface layer of a working part makes it possible to locally refine its surface.

## Figures and Tables

**Figure 1 materials-17-02095-f001:**
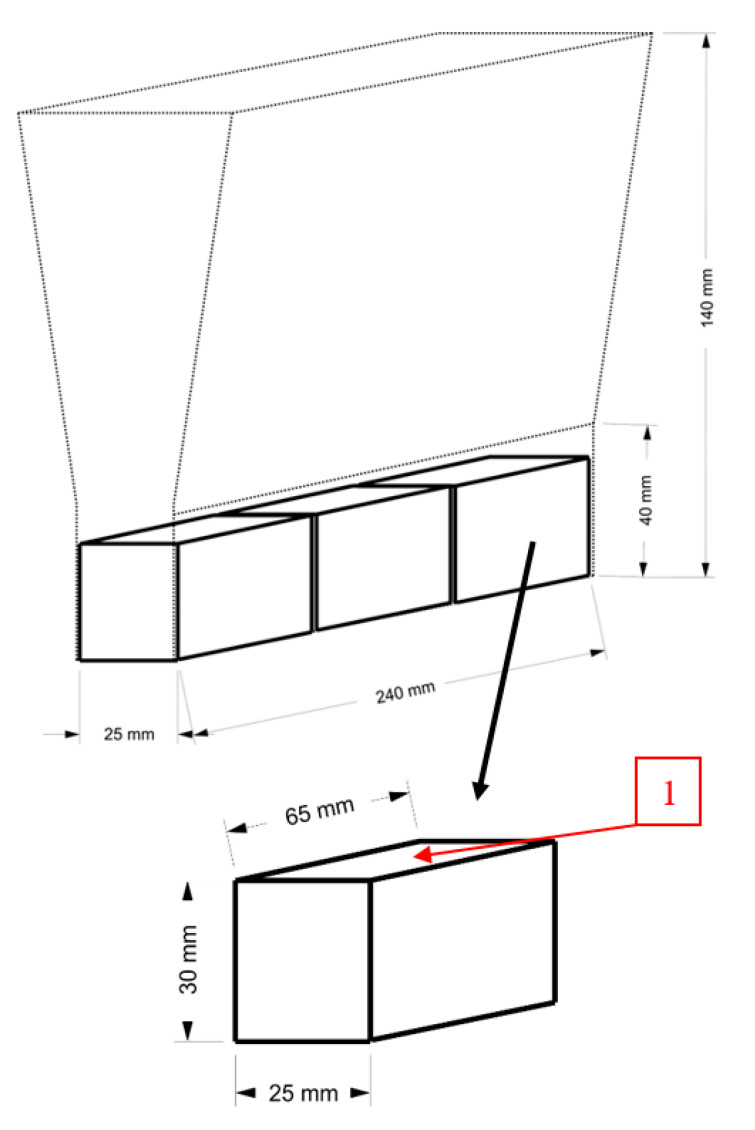
Test ingot and test sample; 1—laser remelted surfaces.

**Figure 2 materials-17-02095-f002:**
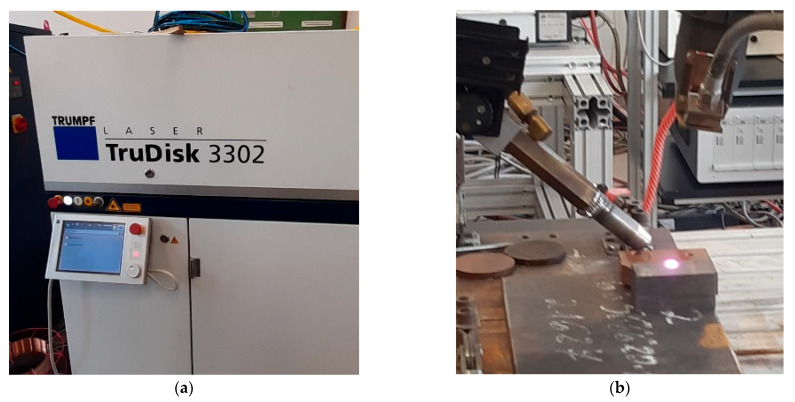
The disc laser used in tests: (**a**) control panel, (**b**) remelting process of sample surface.

**Figure 3 materials-17-02095-f003:**
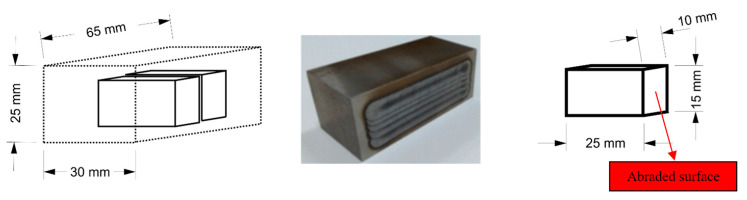
Abrasive wear test sample and the method of collecting it from the nodular cast iron casting after the surface laser remelting.

**Figure 4 materials-17-02095-f004:**
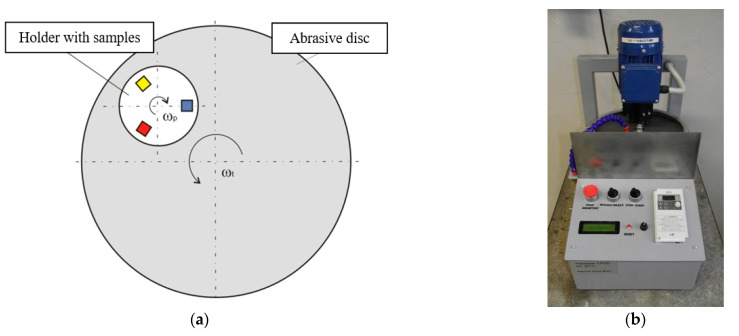
The diagram of the machine operation (**a**) and the general view of the Tribotester 3-POD (**b**).

**Figure 5 materials-17-02095-f005:**
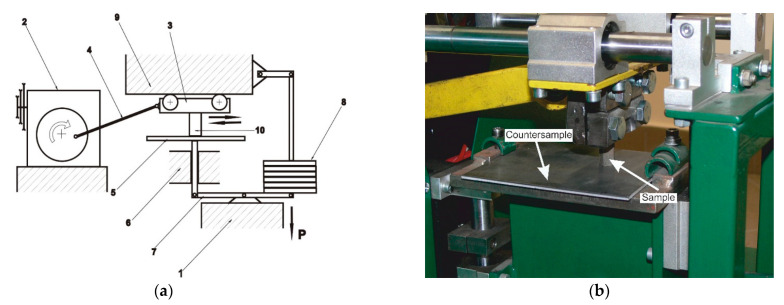
Stand kinematic diagram (**a**) and the general view of the machine (**b**): 1—load-bearing frame, 2—reducer, 3—slider with a chuck, 4—connecting rod, 5—counter-sample, 6—set of counter-sample guides, 7—lever loading system, 8—set of weights, 9—set of slide guides, 10—sample.

**Figure 6 materials-17-02095-f006:**
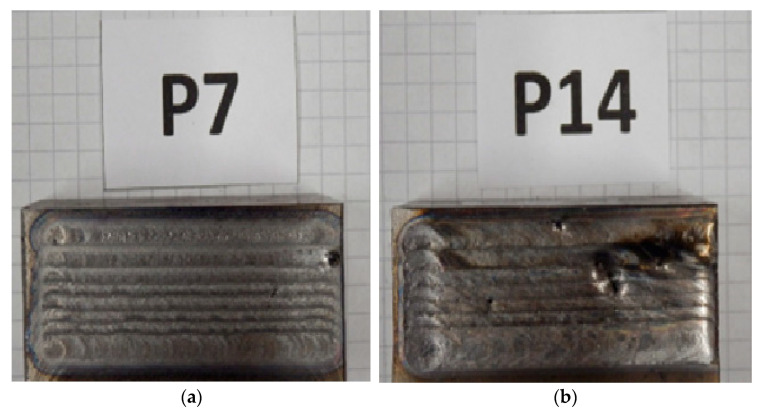
Views of selected sample surfaces after laser remelting: (**a**) sample no. P7, (**b**) sample no. P14.

**Figure 7 materials-17-02095-f007:**
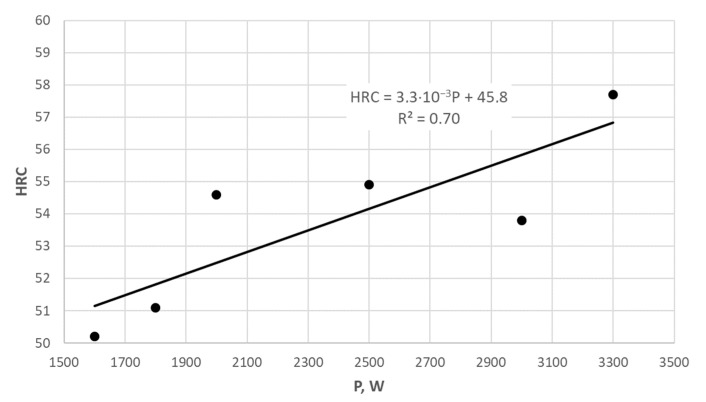
Dependence of hardness HRC on nodular cast iron surface from laser power P at a constant feed speed of 1000 mm/min.

**Figure 8 materials-17-02095-f008:**
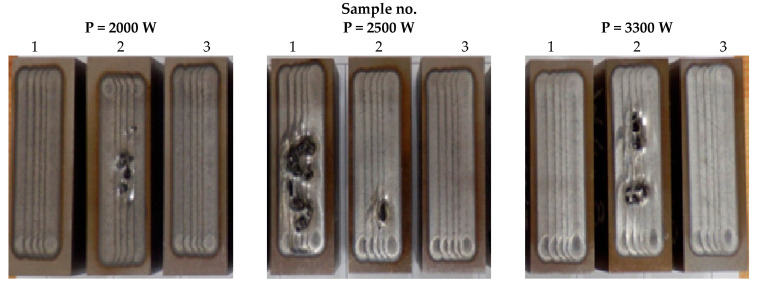
The surface of the example samples (melt W4) after remelting with different laser power.

**Figure 9 materials-17-02095-f009:**
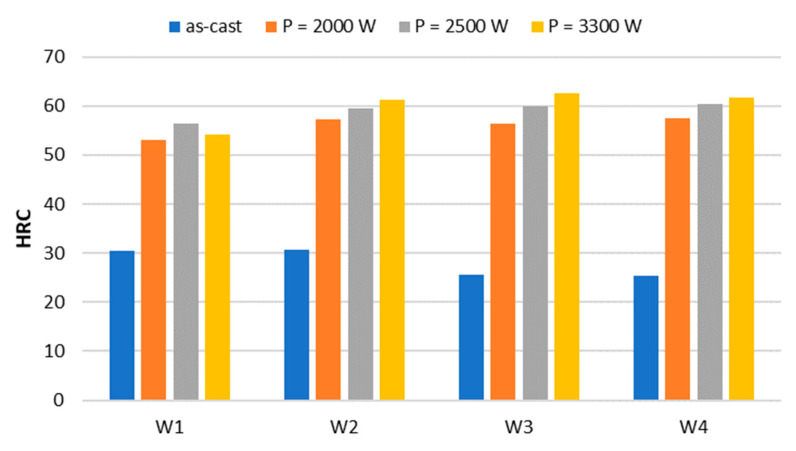
The hardness of the sample surfaces in the as-cast state and after laser remelting.

**Figure 10 materials-17-02095-f010:**
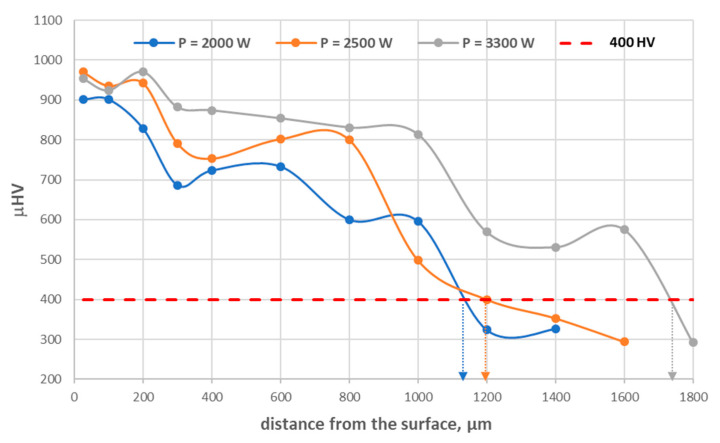
The distribution of microhardness on the W2 sample’s cross-section at different laser powers.

**Figure 11 materials-17-02095-f011:**
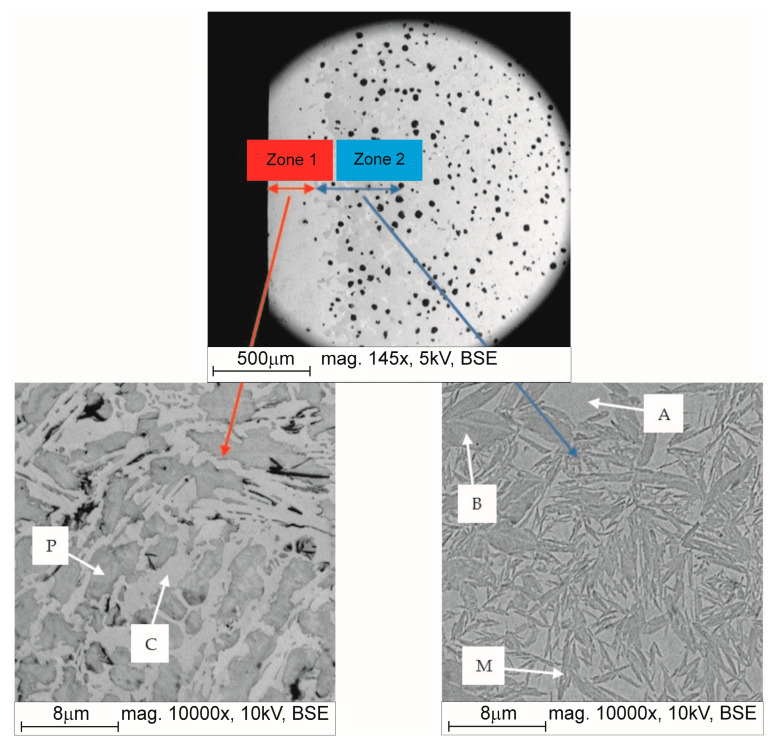
Microstructures of the hardened layer using the laser method on the microsection of sample W2_2500: dendritic microstructure of ledeburite (eutectic mixture of pearlite (P) and cementite (C)) in Zone 1 (left) and acicular martensitic (M) and bainitic (B) microstructure with austenite (A) in Zone 2 (right), SEM, mag. 10,000×.

**Figure 12 materials-17-02095-f012:**
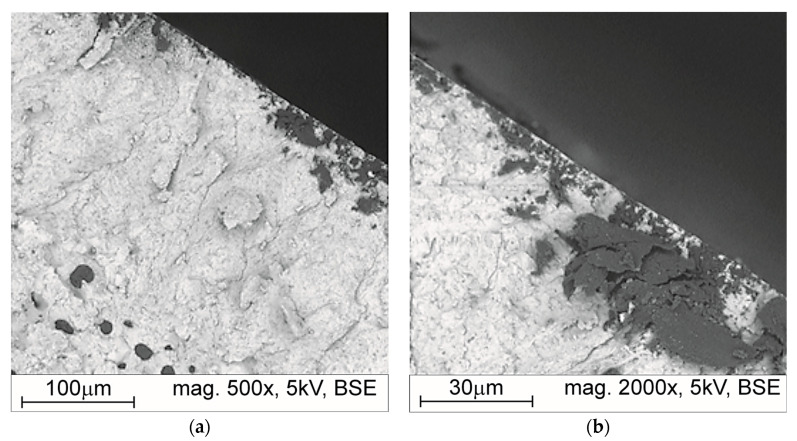
Example fractures of the hardened layer using the laser method on the microsection of sample W2_2500: (**a**) SEM, mag. 500×, (**b**) SEM, mag. 2000×.

**Figure 13 materials-17-02095-f013:**
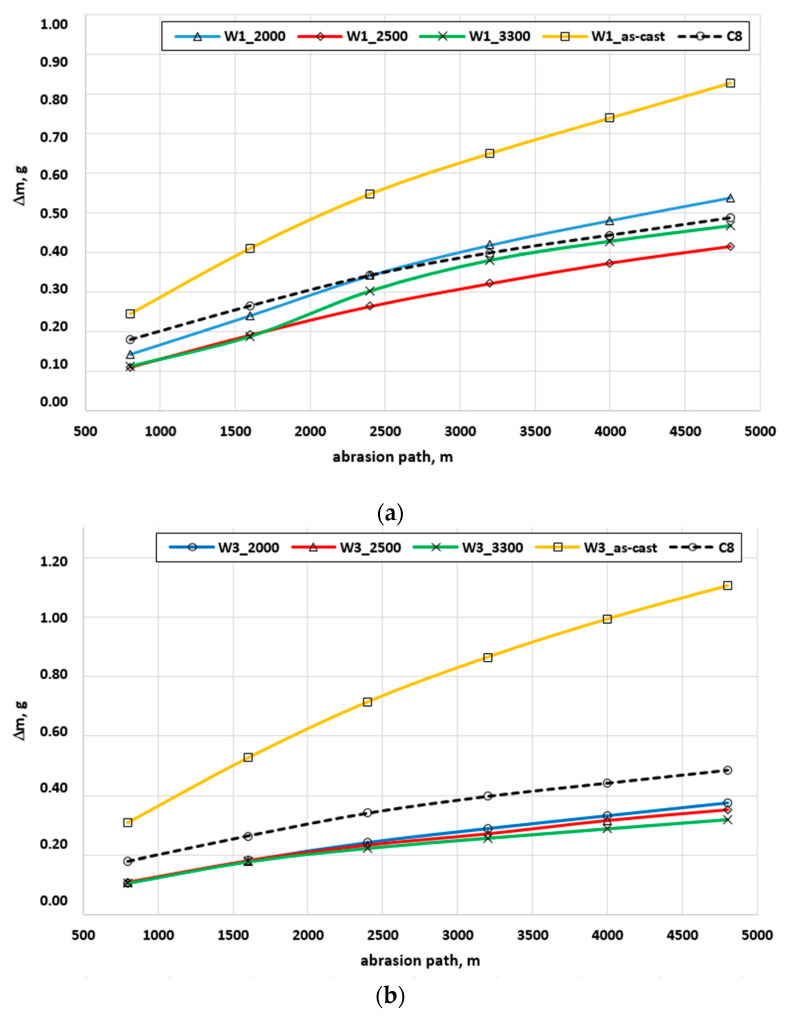
Diagrams of abrasive wear as a function of the abrasion path of the samples for cast iron melts W1 (**a**) and W3 (**b**) in the as-cast state, after laser remelting, and the C8 reference in the tribological system: metal–abrasive paper.

**Figure 14 materials-17-02095-f014:**
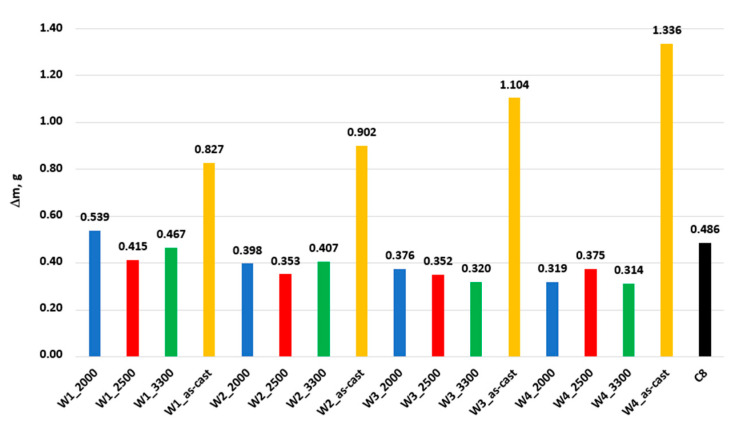
The total wear of the tested samples of nodular cast iron from W1 to W4 in the as-cast state, after the surface laser remelting, and the C8 reference sample in the tribological system: metal–abrasive paper.

**Figure 15 materials-17-02095-f015:**
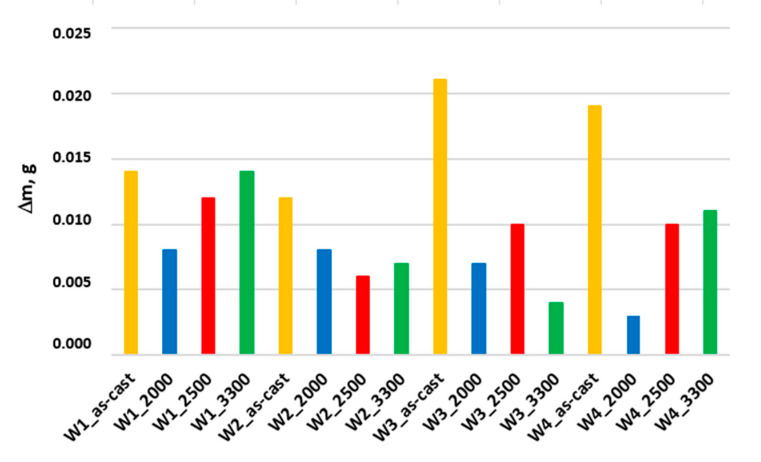
The total wear of the tested samples of nodular cast iron from W1 to W4 in the as-cast state, after surface laser remelting in the tribological system: metal-to-metal.

**Table 1 materials-17-02095-t001:** Selected properties of the cast irons tested in the as-cast state.

Melt	Grade	HB	UTS, MPa	EL, %
W1	GJS-700-4	269	803	4
W2	GJS-750-2	283	777	2
W3	GJS-600-6	283	649	6
W4	GJS-650-3	292	717	3

**Table 2 materials-17-02095-t002:** Laser operating parameters and the obtained hardness of the remelted sample surface.

Sample No.	Laser Power P, W	Beam Feed Rate V, mm/min	The Average Hardness of the Laser-Remelted Sample Surface, HRC
P1	800	600	42.5
P2	800	400	49.2
P3	1000	400	52.9
P4	1200	400	54.3
P5	1200	600	51.7
P6	1400	600	50.1
P7	1400	800	49.1
P8	1600	800	51.6
P9	1600	1000	50.2
P10	1800	1000	51.1
P11	2000	1000	54.6
P12	2500	1000	54.9
P13	3000	1000	53.8
P14	3300	1000	57.7

**Table 3 materials-17-02095-t003:** Average surface hardness of the samples in the as-cast state and after laser remelting.

P, W	HRC			
	W1	W2	W3	W4
as-cast	30.4	30.8	25.6	25.4
2000	53.0	57.3	56.5	57.6
2500	56.3	59.4	60.0	60.3
3300	54.3	61.2	62.5	61.7

**Table 4 materials-17-02095-t004:** The results of microhardness measurements of the tested nodular cast iron after laser treatment.

Sample No.	Distance from the Surface, µm												g, µm
	25	100	200	300	400	600	800	1000	1200	1400	1600	1800	
W1_2000	849	750	707	604	605	680	625	370	344		-	-	950
W1_2500	945	878	707	710	677	700	617	425	290	340	-	-	1050
W1_3300	937	958	960	935	874	630	725	688	692	353	-	-	1350
W2_2000	901	901	828	686	723	733	600	596	324	326	-	-	1150
W2_2500	970	935	942	791	753	802	800	498	400	353	294	-	1200
W2_3300	953	924	970	882	874	854	831	814	569	531	575	292	1750
W3_2000	911	936	779	674	668	694	481	326	280	-	-	-	900
W3_2500	954	951	937	967	682	766	585	585	210	-	-	-	1100
W3_3300	962	950	970	956	952	891	623	618	457	370	276	-	1300
W4_2000	942	965	971	812	705	498	336	325		-	-	-	700
W4_2500	962	991	964	938	911	775	442	443	325	320	-	-	1100
W4_3300	948	928	941	899	897	746	788	657	502	297	262	-	1300

**Table 5 materials-17-02095-t005:** The total weight loss of the samples after the laser treatment of the nodular cast iron castings’ surfaces and a reduction in the sample wear after this treatment as compared to the as-cast state of the sample (in the tribological system: metal–abrasive paper).

Sample No.	Δm, g	Percentage Wear Reduction ^1^, %
W1 (as-cast state)	0.827	-
W1_2000	0.539	35
W1_2500	0.415	50
W1_3300	0.467	43
W2 (as-cast state)	0.902	-
W2_2000	0.398	56
W2_2500	0.353	61
W2_3300	0.407	55
W3 (as-cast state)	1.104	-
W3_2000	0.376	66
W3_2500	0.352	68
W3_3300	0.32	71
W4 (as-cast state)	1.336	-
W4_2000	0.319	76
W4_2500	0.375	72
W4_3300	0.314	76

^1^ cast iron wear reduction compared to the as-cast cast iron state.

**Table 6 materials-17-02095-t006:** The total weight loss of the samples after the laser treatment of the nodular cast iron castings’ surfaces and a reduction in the sample wear after this treatment as compared to the as-cast state of the sample (in the tribological system: metal-to-metal).

Sample No.	Δm, g	Percentage Wear Reduction ^1^, %
W1 (as-cast state)	0.014	-
W1_2000	0.008	43
W1_2500	0.012	14
W1_3300	0.014	0
W2 (as-cast state)	0.012	-
W2_2000	0.008	33
W2_2500	0.006	50
W2_3300	0.007	42
W3 (as-cast state)	0.021	-
W3_2000	0.007	67
W3_2500	0.01	52
W3_3300	0.004	81
W4 (as-cast state)	0.019	-
W4_2000	0.003	84
W4_2500	0.01	47
W4_3300	0.011	42

^1^ cast iron wear reduction compared to the as-cast cast iron state.

## Data Availability

Data are contained within the article.

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
