# Peer review of "Improving the Abrasion Resistance of Nodular Cast Iron Castings by Remelting Their Surfaces by Laser Beam"

_materials, 2024, doi:10.3390/ma17092095_

Round 1

Reviewer 1 Report

Comments and Suggestions for Authors

This is an interesting paper concentrating on the enhancement of abrasion resistance of cast iron by surface remelting. There is an absence of an adequate description of the microhardness tester used and important details about these measurements. Other minor:

Page 2, Line 50: replace show with shown

In general, the use of commas in values/numbers to indicate thousands should be avoided.

Comments on the Quality of English Language

Generally ok, but it could be useful to revisit and make sentences shorter and clearer.

Author Response

Dear Sir,

thank you for Your review of our manuscript materials-2956735 entitled „Improving the abrasion resistance of nodular cast iron castings by remelting their surfaces by laser beam”. All Your comments are addressed in the attached PDF file as suitable responses for Reviewer #1. The necessary modifications made in the manuscript are marked in yellow text (done correction in the range of English is not marked).

Best Regards

Authors of paper

Reviewer 2 Report

Comments and Suggestions for Authors

In this work, the effect of laser hardening treatment on the surface hardness and abrasive wear performance of cast iron has been studied, typically focusing on the laser treatment parameters. However, after review, the current version of the manuscript is not recommended for publication.  

General comments:

1.     Please reorganize the manuscript into different sections. It should generally include Introduction, Experimental methods, Results, Discussion, and Conclusions.

2.     In academic papers, it’s important for authors to provide thorough descriptions and analyses of the results. However, Figures and Tables in the manuscript are not adequately addressed in Lines 241-289, and their interpretation needs to be added.

Some detailed comments are provided below:

1.     Lines 67-102, please properly integrate the purpose and motivation of the study into the last paragraph of the Introduction. Any other content should be related to the experimental methods section.

2.     In Fig. 1., two red arrows highlight two adjacent surfaces. Are these surfaces intended for laser treatment? If so, the boundary regions of these two surfaces may potentially interact with each other. Why did the authors select adjacent surfaces for treatment? 

3.     Lines 104-128, 202-240 and 258-274, which are related to experimental methods, should be moved to the Experimental section.

4.     In Table 2, was the hardness measured directly from the surface of the laser remelted sample or after surface polishing? How to eliminate the effect of rough surfaces on hardness results if the surfaces were not polished.

5.     Line 127, “P4” sample is not at 1000 mm/min. Is it “P9 to P14”? In addition, the hardness of the P13 sample is not higher than that of P11 and P12, suggesting a non-linear trend with laser power. Please clarify it.

6.     Lines 131-134 and Figure 3, (1) P7 sample was not treated under the highest laser power, which is not consistent with the statement. (2) where is “the surface of the samples prior to treatment (after machining)”?

7.     Line 141, the statement “It was decided that the feed speed would be as high as possible, i.e. 1,000” is contradicted by the statement “so for samples P1 to P4 a significant increase in hardness can be seen with increasing laser power and lowering the feed rate, which results in stronger surface remelting” as well as Table 2. Why is the high feed speed expected? (2) Line 142, Why choose 2,000, 2,500 and 3,000 W? These laser powers appear to be more costly, from the economic point of view for laser treatment (as stated in Line 124)

8.     Table 3, are all samples W1? Please revise.

9.     Figure 6 shows that the overall trend of hardness decreases with increasing distance from the surface. However, rather than following a linear trajectory, this is a waveform change. Why?

10.  Lines 165-170, please add relevant references. For Figure 7, please add appropriate labels to describe the microstructure.

11.  Lines 255 and 286, both of them are table 5?

12.  Line 321-323, based on Table 5 (Line 255), the wear reduction is obvious compared to the as-cast state. Why do the authors state that “the laser melting method do not significantly increase the abrasion resistance in the metal-ceramic”?

13.  In this work, the weight loss method was used to assess wear resistance. However, during testing, adhesive wear could occur, affecting the final weight change results due to the transfer of material to the sample surface. How to eliminate this situation?

Comments on the Quality of English Language

Minor editing of English language is suggested

Author Response

Dear Sir,

thank you for Your review of our manuscript materials-2956735 entitled „Improving the abrasion resistance of nodular cast iron castings by remelting their surfaces by laser beam”. All Your comments are addressed in the attached PDF file as suitable responses for Reviewer #2. The necessary modifications made in the manuscript are marked in yellow text (done correction in the range of English is not marked).

Best Regards

Authors of paper

Reviewer 3 Report

Comments and Suggestions for Authors

The authors present results of an extensive research on influence of the laser beam remelting of the nodular cast iron surfaces on the abrasion resistance of the material. The results of experimental work on the two types of the nodular cast iron include measuring the materials’ hardness and studying the microstructure, as well as the tribological pin-on-disc test with abrasive paper and unalloyed steel. The paper also contains well documented discussion of those results and the conclusions that followed from it.

There are two questions that might be of interest to be resolved. First, (page 2 – lines 78 to 83) why the materials composition is being kept as a secret since if it is already submitted for patenting, so those compositions should be considered as protected. Thus, besides the presented results, authors would have been in the position to compare results of their investigations for the two different materials, i.e., is there any influence of laser melting that can be related to different chemical compositions of the two materials.

The second point is the setup of the tribometer test, namely explanation of figure 10 (page 10 – lines 208 to 211). If both the disc as well as the samples’ holder are rotating, would it be reasonable to assume that trajectories of the three samples would eventually cross, and samples would not have the same surface conditions, especially if the different pressures are applied to each of them? That might not be of great importance when the disc is of the hard abrasive paper, but for the second case, when the disc is relatively softer metal, it would make a difference.

Page 13 – lines 272-273 – “Before each test cycle, the scales were checked  with mass standards”. Does this mean that the calibration of the scales was done for each measurement? Please, explain and use proper terminology.

There are several paragraphs highlighted in yellow, which would mean that the authors did some changes and that the final version of the manuscript is not submitted to the journal?

The scanned pages of the manuscript with marked errors and suggested corrections are enclosed.

Comments on the Quality of English Language

Minor problems with determinate articles.

Author Response

Dear Sir,

thank you for Your review of our manuscript materials-2956735 entitled „Improving the abrasion resistance of nodular cast iron castings by remelting their surfaces by laser beam”. All Your comments are addressed in the attached PDF file as suitable responses for Reviewer #3. The necessary modifications made in the manuscript are marked in yellow text (done correction in the range of English is not marked).

Best Regards

Authors of paper

Round 2

Reviewer 2 Report

Comments and Suggestions for Authors

No further comments on this work.